# Increased Phenoloxidase Activity Constitutes the Main Defense Strategy of *Trichoplusia ni* Larvae against Fungal Entomopathogenic Infections

**DOI:** 10.3390/insects14080667

**Published:** 2023-07-25

**Authors:** Kristin R. Duffield, Alayna M. Rosales, Ephantus J. Muturi, Robert W. Behle, José L. Ramirez

**Affiliations:** 1USDA-ARS, National Center for Agricultural Utilization Research, Crop BioProtection Research Unit, 1815 N. University St., Peoria, IL 61604, USA; ephantus.muturi@usda.gov (E.J.M.);; 2Biology Department, Bradley University, Peoria, IL 61604, USA; amrosales@mail.bradley.edu

**Keywords:** entomopathogenic fungi (EPF), host immune response, host–fungal interactions, phenoloxidase cascade, insect immunity

## Abstract

**Simple Summary:**

The cabbage looper is an important agricultural pest worldwide and is frequently used as a model organism for measuring the effectiveness of biopesticides, including those derived from pesticidal fungi. To better understand how cabbage loopers resist and survive infection of insect-killing fungi, we measured several insect defense parameters following exposure to two species of fungi. Our results indicate that the cabbage looper’s main defense mechanism is the melanization cascade, with activation of several melanization-related genes soon after infection. This differs from what has been reported in other insect–fungi studies and might represent a target for disruption in future cabbage looper control strategies that integrate fungal biopesticides. Our study also shows that one well-known insect antifungal immune response does not participate in the cabbage looper antifungal defense, differentiating this important pest from other insects. Overall, this study provides insights into the host response strategies employed by cabbage loopers for protection against pesticidal fungi and aids in the design of more effective biological control strategies for this pest.

**Abstract:**

The cabbage looper *Trichoplusia ni* is an important agricultural pest worldwide and is frequently used as a model organism for assessing entomopathogenic fungi virulence, though few studies have measured the host response repertoire to fungal biocontrol agents. Here, we quantified the immune response of *T. ni* larvae following exposure to two entomopathogenic fungal species: *Beauveria bassiana* and *Cordyceps javanica*. Results from our study demonstrate that *T. ni* larvae exposed to fungal entomopathogens had higher total phenoloxidase activity compared to controls, indicating that the melanization cascade is one of the main immune components driving defense against fungal infection and contrasting observations from other insect–fungi interaction studies. We also observed differences in host response depending on the species of entomopathogenic fungi, with significantly higher induction observed during infections with *B. bassiana* than with *C. javanica*. Larvae exposed to *B. bassiana* had an increased expression of genes involved in prophenoloxidase response and the Imd, JNK, and Jak/STAT immune signaling pathways. Our results indicate a notable absence of Toll pathway-related responses, further contrasting results to other insect–fungi pathosystems. Important differences were also observed in the induction of antimicrobial effectors, with *B. bassiana* infections eliciting three antimicrobial effectors (lysozyme, gloverin, and cecropin), while *C. javanica* only induced cecropin expression. These results provide insight into the host response strategies employed by *T. ni* for protection against entomopathogenic fungi and increase our understanding of insect–fungal entomopathogen interactions, aiding in the design of more effective microbial control strategies for this important agricultural pest.

## 1. Introduction

Entomopathogenic fungi abound in nature, where they act as natural regulators of insect populations [1,2]. Due to their efficacy, mode of direct contact transmission, and range of host specificity, fungal pathogens have been used as biological control agents against insect pests for centuries. Since the 1970s, the development of commercialized products utilizing entomopathogenic fungi (e.g., from *Metarhizium*, *Beauveria*, and *Cordyceps* genera) as environmentally friendly alternatives to traditional chemical pesticides has greatly expanded [3,4]. Importantly, the success of fungal pathogens depends on both host (e.g., physical state, resistance) and environmental factors (e.g., temperature, humidity, UV exposure) [5], which can impose severe limitations on their use as a cost-effective biological control [6]. Moreover, compared to chemical pesticides, fungal pathogens have lower virulence (i.e., slow kill speed with a higher inoculum required). Recent research efforts have considerably improved fungal biocontrol agents [7,8]. For example, genetic modifications have resulted in fungal strains with increased virulence and resistance to abiotic stress [9,10].

Entomopathogenic fungi are typically screened in agricultural pests via bioassays that measure host mortality, and often mycosis, across increasing doses (e.g., survival analysis) [11]. However, little consideration is given to evaluating insect host responses following exposure to fungal pathogens despite their role in determining infection outcomes. Consequently, we know far less about the host response of many agricultural insect pests to entomopathogenic fungi [12]. Understanding the host response (i.e., immunity) of target pests to entomopathogenic fungi may improve the modification, selection, and application of mycoinsecticides, as characterizing immune responses across groups could reveal variables of host susceptibility. For example, solitary migratory locusts, *Locusta migratoria,* were found to be less susceptible to *Metarhizium anisopliae* infection than gregarious migratory locusts. Transcriptome analysis of these two populations revealed that locusts from gregarious populations were prophylactically upregulating immune function, which likely conferred greater protection [13].

Insects possess a suite of countermeasures to fight and defend against infection by fungal pathogens, and populations can adapt their responses over time [14] and within generations via immune priming [15]. Most of what is known about the molecular basis of the insect immune response to fungal infections comes from work in *Drosophila melanogaster* [16,17] and some lepidopteran [18] and mosquito species [19,20,21]. For infection to establish, fungal spores (or conidia) must attach to and penetrate the insect cuticle via a combination of enzymes, after which fungal blastospores bud off hyphae, multiply, and circulate within the insect hemocoel. There, fungal cells are detected via pattern recognition receptors (PRRs), which quickly activate innate immune defenses, including both humoral and cellular immunity [22]. Within the humoral defense, the Toll pathway is the primary immune signaling pathway to respond to fungi in *Drosophila* [23]. Additionally, the immunodeficiency (Imd), Janus kinase-signal transducer and activator of transcription (Jak-STAT), and c-Jun N-terminal kinase (JNK) pathways play important roles in insect defense against pathogens, including fungi [24]. Upon pathogen recognition, signals then pass through intracellular signal transduction cascades, leading to the expression of potent antimicrobial effectors that are primarily produced in the fat bodies but also by epidermis cells and hemocytes [25]. Lepidopterans are known to produce several classes of antimicrobial effectors, such as lysozymes, cecropins, gloverins [26], and defensins [27], including gallerimycin [28]. While the role that specific antimicrobial effectors play in entomopathogenic fungi defense is largely unknown, there are some reported examples. For instance, cecropin A and gloverin2 were found to have high antifungal activity against *Beauveria bassiana* in silkworm larvae (*Bombyx mori*) [29,30].

The prophenoloxidase (proPO) activation system is another component of the insect humoral immune response, which regulates the coagulation and melanization of hemolymph in response to pathogens. It comprises both short (e.g., reactive quinones) and long-lasting (e.g., melanin) products that are quickly deposited to surround invading fungi to encapsulate and kill [31]. Mounting a potent immune response not only attacks pathogens but can also cause collateral damage to host cells. As such, antioxidant enzymes are deployed following exposure to biotic and abiotic stressors, including pathogens, to protect host cells from Reactive Oxygen Species (ROS).

The cabbage looper *Trichoplusia ni* (Lepidoptera: Noctuidae) is a common agricultural pest throughout much of the world and has long been used as a representative lepidopteran for studying insect physiology [32,33,34,35], including insect pathology [36,37,38,39]. Furthermore, *T. ni* has frequently been used as a model insect for assessing the efficacy of biopesticides [40,41,42,43,44] as well as resistance evolution, as *T. ni* is known to have developed resistance to *Bacillus thuringiensis* (*Bt*), a widely used microbial insecticide [45,46]. However, less is known about the immune response of *T. ni* to fungal entomopathogens, especially at the transcription level. In the present study, we characterized the immune response of *T. ni* larvae to two entomopathogenic fungal species, *B. bassiana* (MBC 076) and *Cordyceps javanica* (MBC 439) (formerly *Isaria javanica*). Both fungal species are known to infect a broad insect host range [47], including lepidopterans. Here, we quantified the total phenoloxidase (PO) activity in addition to measuring the expression of immune gene markers across Toll, Imd, Jak/STAT, and JNK immune signaling pathways; propohenoloxidase (proPO) response; and stress and detoxification response using real-time PCR (qPCR).

## 2. Materials and Methods

### 2.1. Insect Rearing

We assessed immune responses following fungal exposure in 5-day-old cabbage looper larvae. Larvae were reared in individual cups in the dark on ‘general purpose lepidoptera diet’ (Frontier Agricultural Sciences, Newark, DE, USA) at 27 ± 2 °C and 55 ± 10% relative humidity until fungal exposure at 5 days old.

### 2.2. Entomopathogenic Fungi Exposure

Larvae were exposed to two common entomopathogenic fungal strains: *Beauveria bassiana* (MBC 076) and *Cordyceps javanica* (MBC 439). Fungal cultures were conducted as previously described [21]. Briefly, cultures of *B. bassiana* (MBC 076) and *C. javanica* (MBC 439) were grown on ¼ strength Sabouraud dextrose agar and yeast extract (SDAY) medium at 26 °C for 15 days. Conidia were harvested from plates with a sterile inoculation loop and suspended in 3 mL of 0.04% Tween-20 (Invitrogen, ThermoFisher Scientific, Waltham, MA, USA). To remove mycelia, conidia were filtered using a syringe filter (1 mL) which were then used to make 1 × 10^8^ concentrations for inoculation.

For fungal inoculation, ten larvae were placed into small cups containing 2 mL of their respective treatment (either 1 × 10^8^
*B. bassiana* conidia in 0.04% Tween-20, 1 × 10^8^
*C. javanica* conidia in 0.04% Tween-20, or 0.04% Tween-20 for control larvae). Larvae were gently swirled for 20 s and then carefully removed with wide-tip entomology forceps and placed on a dry paper towel before being returned to their original rearing environment.

### 2.3. Phenoloxidase Activity Assay

The melanization cascade is often an important antifungal defense within insects. Here, we measured total phenoloxidase (PO) activity as described in Sadd et al. [48], with some modifications. Briefly, *T. ni* larvae that have undergone fungal exposure were transferred to their original rearing cup and maintained under rearing conditions described above. At 2 days post-infection, larvae were cold anesthetized on ice, and their back proleg was cut with sterile micro scissors to collect 10 µL of outflowing hemolymph. The collected hemolymph was mixed with 40 µL of cold 1× PBS, flash-frozen with liquid nitrogen, and stored at −80 °C. PO assays were conducted with 15 µL of sample (hemolymph in 1× PBS) or 15 µL of 1× PBS (negative control). Samples were transferred to a flat-bottomed 96-well plate containing 140 µL of molecular-grade water and 20 µL 1× PBS. A 20 µL L-Dopa suspension (4 mg/mL H2); 3,4 dihyroxy-L-phenylalanine was then added to each well and mixed for 5 s at 30 °C in a spectrophotometer (Multiskan GO, Thermo Scientific, Waltham, MA, USA). The absorbance was read at 490 nm every 15 s with 5 s of shaking between reads. PO activity was calculated from the slope (V_max_) of the reaction curve during its linear phase over 160 readings. Each sample was read in duplicate and the average was used to evaluate statistical significance. Three independent experiments were conducted with 10 individuals per treatment and each experiment was conducted with a fresh batch of larvae and fungal spores.

### 2.4. Reverse Transcriptase Quantitative PCR (RT-qPCR) Detection and Quantification

To assess immune function at the transcription level, we performed gene expression analysis of genes across multiple immune pathways. Here, larvae were killed 2 days post-infection (sample size = 13 larvae per treatment) by freezing. RNA was extracted from individual whole-bodied larvae using TRIzol (Invitrogen) according to the manufacturer’s instructions. Concentration and quality of the extracted RNA were evaluated via NanoDrop (Thermo Scientific, Waltham, MA, USA). Synthesis of cDNA was conducted on normalized amounts of RNA using the QuantiTec reverse transcription kit with DNA Wipeout (Qiagen, Hilden, Germany).

To assess a range of potential genes involved in the antifungal immune response, we targeted multiple genes across different facets of insect immunity, including those involved in the prophenoloxidase (proPO) response as well as the Toll, Imd, Jak/STAT, and JNK immune signaling pathways and the stress and detoxification response. We screened several previously established gene targets of *T. ni* predicted to be involved in immune function [37,38] We also designed several primers targeting key immune genes (Table 1) using Primer3 (v 0.4.0) [49,50] based on other previous studies [51,52]. Fungal and bacterial load was quantified by measuring the transcript abundance of the 18s rRNA gene using fungal-specific primers designed to amplify a highly conserved region among fungi [53] and the 16s rRNA gene using bacterial-specific primers designed to amplify a highly conserved region among bacteria (all primers from IDT, Inc., Newark, NJ, USA).

We used a primer targeting the mitochondrial ribosomal protein S18 (Table 1) as our reference gene target throughout for quantification purposes. This gene was selected based on its superior performance/ranking to other genes via RefFinder [54,55], which uses the algorithm from major computational programs such as geNorm, Normfinder, and Best-Keeper to compare and rank candidate reference genes. Prior to conducting qPCR assays, RNA was normalized to 1 µg, treated with DNA Wipeout, and then converted to cDNA using the QuantiTect Reverse Transcription Kit (Qiagen, Hilden, Germany).

Real-time qPCR reactions were run on a Quant-Studio 6 Real-Time PCR instrument (Thermo Fisher Scientific, Waltham, MA, USA), and included a melt-curve stage to confirm product specificity. One microliter of cDNA product was used in a 10 µL qPCR reaction using gene-specific primers (Table 1) and PowerUp SYBR Green Master Mix kit (Qiagen). qPCR cycling conditions consisted of holding at 50 °C for 2 min and 95 °C for 2 min and 40 cycles of 1 s at 95 °C and 30 min at 60 °C. Gene expression profiles were evaluated post-run using the ΔΔCt method [56].

### 2.5. Survival

To measure the impact of our fungal treatments on survival, mortality was measured daily following fungal treatment inoculation until pupation (*n* = 30 per treatment). In a separate group of larvae, if an individual survived to pupation, they were included as censored values in the subsequent analysis.

### 2.6. Statistical Analysis

All statistical analyses were conducted using GraphPad Prism 9 (Version 9.4.1). For expression data, we first removed outliers via ROUT using default settings (Q = 1%) [57]. To fit assumptions of normality, expression data for most target genes were log^2^ transformed. The exceptions to this were for expression of *Gallerimycin*, which was square-root transformed, as well as PO activity, *STAT5B*, *Defensin*, *JNK*, *JUN*, and fungal 16s rRNA; all of which were not transformed.

For most tests, we analyzed the expression data and PO activity with a standard one-way ANOVA with Dunnett’s multiple comparisons test. However, due to unequal standard deviations in some cases, we employed a Welch ANOVA with Dunnett’s multiple comparisons test for expression of the genes *PPO2*, *Cactus-like*, *Relish*, *JNK*, and *Gloverin*. Expression data for *STAT5B* and fungal 18s rRNA were analyzed with the nonparametric Kruskal–Wallis test with Dunn’s multiple comparisons test. Larval survival to pupation was analyzed via a Kaplan–Meier with Log-rank test. For all tests, statistical significance was assessed at *p* < 0.05, with the strength of the significance represented with asterisks (* *p* < 0.05; ** *p* < 0.01; *** *p* < 0.001; **** *p* < 0.0001).

## 3. Results

### 3.1. Survival

In a subset of *T. ni* larvae, we quantified survival rate to pupation following exposure to sublethal doses of *B. bassiana* and *C. javanica*. Mortality was very low across all larvae. Within each treatment group, two out of thirty (6.7%) larvae died prior to pupal molt, and neither exposure to *C. javanica* nor *B. bassiana* increased mortality in *T. ni* larvae (Mantel-Cox: χ^2^_2_ = 0.001; *p* = 0.9993).

### 3.2. Microbial Loads

We evaluated the total fungal and bacterial load in *T. ni* across fungal treatments by quantifying the expression of fungal 18s rRNA or bacterial 16s rRNA, respectively. Our fungal treatments significantly affected total fungal loads within *T. ni* larvae (Kruskal–Wallis test: *H* = 10.69, *p* = 0.0048; Figure 1). Specifically, both *T. ni* larvae exposed to *C. javanica* (Dunn’s: *p* = 0.0024) and *B. bassiana* (Dunn’s: *p* = 0.0292) had a higher expression of fungal 18s rRNA compared to control larvae. For bacterial 16s rRNA, our main ANOVA model was statistically significant (ANOVA: *F*_2, 36_ = 3.340; *p* = 0.0467); however, neither larvae exposed to *C. javanica* (Dunnett’s: *p* = 0.9352) nor *B. bassiana* (Dunnett’s: *p* = 0.0823) had different total bacterial loads compared to controls (Figure 1).

### 3.3. Total Phenoloxidase (PO) Activity

Using an enzymatic assay, we assessed the phenoloxidase activity following exposure to our fungal treatments. Larvae exposed to either fungal species in this study exhibited significantly higher PO activity 2 days post-exposure compared to controls (ANOVA: *F*_2, 107_ = 20.30; *p* < 0.0001; Dunnett’s: *B. bassiana p* < 0.0001; *C. javanica p* = 0.0077; Figure 2A).

### 3.4. Gene Expression

We evaluated the relative expression of genes involved in the prophenoloxidase response, immune signaling pathways (Toll, Imd, JNK, and Jak/STAT), multiple antimicrobial effectors (*Cecropin*, *Defensin*, *Lysozyme*, *Gallerimycin*, and *Gloverin*), and *Catalase*, which plays a role in detoxification of ROS as well as the host stress response.

#### 3.4.1. Prophenoloxidase (proPO)

Multiple genes involved in the prophenoloxidase response of *T. ni* larvae were induced by inoculation of our fungal pathogen treatments (ANOVA: *PPOAE*: *F*_2, 34_ = 7.596; *p* = 0.0019; *PPO2*: *W*_2, 21.29_ = 7.664; *p* = 0.0031; Figure 2B). Specifically, *T. ni* larvae exposed to *B. bassiana* had a higher expression of both ProPO activating enzyme (*PPOAE*) and ProPO subunit 2 (*PPO2*) (Dunnett’s: *PPOAE*: *p* = 0.0012, *PPO2*: *p* = 0.0016; Figure 2B), while *C. javanica*-treated larvae had similar expression levels to control larvae (Dunnett’s: *PPOAE*: *p* = 0.5221, *PPO2*: *p* = 0.0751; Figure 2B). The expression of ProPO subunit 1 (*PPO1*) was similar across all treatment groups (ANOVA: *F*_2, 36_ = 2.243; *p* = 0.1208; Dunnett’s: *B. bassiana p* = 0.0750; *C. javanica p* = 0.5283; Figure 2B).

#### 3.4.2. Immune Signaling Pathways

Based on our data, neither *B. bassiana* nor *C. javanica* induced gene targets in the Toll immune signaling pathway of *T. ni* larvae 2 days post-infection (Figure 3A). Expression of *Tollo* (ANOVA: *F*_2,36_ = 1.254; *p* = 0.2974), *Toll-7* (ANOVA: *F*_2,36_ = 1.269; *p* = 0.2934), *Cactus-like* (Welch ANOVA: *W*_2,23.15_ = 2.263; *p* = 0.1266), and *Dorsal/Dif* (ANOVA: *F*_2,35_ = 0.4302; *p* = 0.6538) was similar across all groups.

Within the Imd immune signaling pathway, the expression of *Relish* was significantly higher in larvae exposed to *B. bassiana* compared to controls (Welch ANOVA: *W*_2,21.24_ = 3.261; *p* = 0.0582; Dunnett’s: *p* = 0.0373), but similar to controls in larvae exposed to *C. javanica* (Dunnett’s: *p* = 0.5496; Figure 3B). *IKK-β* expression was comparable across all treatments (ANOVA: *F*_2,35_ = 0.2261; *p* = 0.7988; Figure 3B).

*JNK* expression was significantly higher in *B. bassiana*-treated larvae (Welch ANOVA: *W*_2,19.45_ = 3.658; *p* = 0.0448; Dunnett’s: *p* = 0.0272), but similar in control and *C. javanica*-treated larvae (Dunnett’s: *p* = 0.5612; Figure 3C). *JUN*, another target within the JNK immune signaling pathway, was similar across all treatments (ANOVA: *F*_2,35_ = 0.01312; *p* = 0.9870; Figure 3C).

Finally, within the Jak/STAT immune signaling pathway, the expression of *STAT5B* was significantly higher in larvae exposed to *B. bassiana* (Kruskal–Wallis test: *H* = 5.788, *p* = 0.0553; Dunn’s: *p* = 0.0336), while the expression in *C. javanica*-exposed larvae and controls did not differ from each other (Dunn’s: *p* = 0.3068; Figure 4A).

#### 3.4.3. Stress Response-Detoxification

*Catalase* was not differentially expressed across fungal treatments (ANOVA: *F*_2,34_ = 1.903; *p* = 0.1647; Figure 4B).

#### 3.4.4. Antimicrobial Effector Molecules

*Cecropin* was expressed significantly higher in both fungal treatments compared to controls (ANOVA: *F*_2,34_ = 36.29; *p* < 0.0001; Dunnett’s: *B. bassiana p* < 0.0001; *C. javanica p* = 0.0026; Figure 4C). Additionally, larvae exposed to *B. bassiana* had a higher expression of both *Lysozyme* (ANOVA: *F*_2,34_ = 3.197; *p* = 0.0534; Dunnett’s: *p* = 0.0315) and *Gloverin* (Welch ANOVA: *W*_2,20.85_ = 6.672; *p* = 0.0058; Dunnett’s: *p* = 0.0037); however, there was no change in expression between *C. javanica*-exposed larvae and controls (Dunnett’s: *Lysozyme p* = 0.5339; *Gloverin p* = 0.3377; Figure 4C). Finally, the expression of *Defensin* (ANOVA: *F*_2,36_ = 1.572; *p* = 0.2215) and *Gallerimycin* (ANOVA: *F*_2,33_ = 1.660; *p* = 0.2057; Figure 4C) was similar across all groups.

## 4. Discussion

Host-derived responses and differences in fungal strain virulence are known to significantly influence the outcome of fungal entomopathogenic infections in insects. In this study, we evaluated the *T. ni* larvae anti-fungal host response following a challenge with two different species of entomopathogenic fungi.

Our survival bioassays indicated low mortality of *T. ni* larvae exposed to *B. bassiana* (MBC 076) or *C. javanica* (MBC 439). This is similar to what has been previously documented with bioassays that include exposure of second instar *T. ni* larvae to *B. bassiana* [58] but contrasts with findings from other studies [44]. One likely explanation for this is the variation in fungal virulence among fungal strains and the mode of infection used in these studies. Additionally, it could be due to differences in the age of larvae at the time of inoculation. While we chose to infect at 5 days old when larvae were easier to handle without risk of injury, many previous studies have demonstrated significant mortality to neonate larvae [59]. These results suggest that *T. ni* larvae are only susceptible to these fungal strains for a very short time, which could have important implication in field applications. Future studies exploring the effect of host age on fungal susceptibility, in addition to changes in host immune response, would provide critical information on host susceptibility in this model system. Still, fungal loads were higher in both fungal treatment groups compared to controls, suggesting either that fungi were able to proliferate within the host, yielding a sublethal infection, or that they are confined within the cuticle and unable to infect the host. We argue that the former scenario is more likely due to the upregulation of the immunity of inoculated larvae observed in this study.

Our study indicates that the melanization cascade is one of the main components driving the *T. ni* larval defense against fungal infection. We found a significant induction of phenoloxidase activity in larvae infected with both species of fungi, with larvae exposed to *B. bassiana* having significantly higher PO activity in comparison to *C. javanica*, which might indicate a higher virulence of *B. bassiana*. These results are further corroborated by the significant induction of *PPOAE* and *PPO2*, two important prophenoloxidase cascade genes, in *B. bassiana*-infected larvae. A similar induction in PO activity has been observed in the Asian corn borer *Ostrinia furnacalis* during infections with *B. bassiana* [60]. However, this contrasts with the significant decrease in PO activity observed in the greater wax moth *Galleria mellonella*–*B. bassiana* infection system [61]. Prophenoloxidase responses post-fungal infection have been shown to vary according to the host–fungal combination [61,62], and it is most likely determined in part by fungal virulence. For instance, studies conducted by Vertyporokh and Wojda [63] demonstrated strong PO activation in *G. mellonella* larvae injected with sublethal doses of the fungus *Candida albicans* but a significant inhibition with injections of lethal doses. Provided *C. albicans* is not a known entomopathogenic fungus, these findings might indicate a plasticity of the prophenoloxidase cascade response during microbial infections, with the intensity of the response increasing with sublethal doses (i.e., focalized infections) but suppressed when infections are well established or disseminated in the insect body. Phenoloxidase activation and the process of melanization generate highly cytotoxic quinones and other reactive compounds that are designed to control microbial infection [64]. Thus, a potential modulation of the proPO cascade that is dependent on infection level could represent a strategy to overcome the costly and potentially deleterious effect of the overactivation of the prophenoloxidase cascade to host tissues, at a time when other immune-related defenses are much more appropriate or effective. However, further studies evaluating the diverse set of genes and pathways associated with prophenoloxidase control are needed to discern their role in the progression of fungal infections. Fungal infection and dissemination in insects are also restricted by the action of immune signaling pathways, such as Toll, Imd, and Jak/STAT. Surprisingly, the Toll pathway, which is mainly responsible for the recognition and defense against fungi and Gram-positive bacteria in insects, was not induced by any of the two entomopathogenic fungi tested. This differs from its upregulation in domestic silkworm *Bombyx mori* infections with *B. bassiana* [65,66] or those observed in mosquitoes [19]. This difference might be due to either the infecting strain, the sublethal dose of infection, or the characteristic of *T. ni* at this larval stage. Further studies are needed to fully elucidate this pathway during fungal infections in this lepidopteran host.

Further analysis of other important immune pathway components suggests that the antifungal immune repertoire of *T. ni* larvae includes the participation of the Imd, JNK, and Jak/STAT pathways. However, this appears to be fungal-strain-specific, with significant induction following *B. bassiana* infections but less apparent following *C. javanica* infections. Similar infection phenotypes, with inductions of the Imd and JAK/STAT pathways, have been found in *B. mori* during infections with *B. bassiana* [65,67] but to our knowledge, our studies are the first to report their potential implications in the antifungal response of *T. ni* larvae.

The elicitation of immune signaling cascades during microbial infections results in the production of antimicrobial effectors aimed at halting the spread of the infection [16]. In particular, the production of antimicrobial peptides (AMPs) is recognized to be paramount in the insect response to microbial infections. In our study, exposure to both fungal pathogens resulted in the significant induction of the antimicrobial peptide *Cecropin*. In contrast, following the same trend of immune pathway induction, only infections with *B. bassiana* resulted in higher expression of the antimicrobial effectors *Lysozyme* and *Gloverin.* These antimicrobial effectors are also elicited in other lepidopteran hosts; for instance, during *B. bassiana* infection in *B. mori* [29,65], the cotton bollworm *Helicoverpa armigera* [68], and in the fall armyworm *Spodoptera frugiperda* in response to *Metarhizium rileyi* infections [69]. Furthermore, it appears that some AMPs are working in concert to limit fungal infection similar to cecA and gloverin during *B. mori* responses to *B. bassiana* [30]. Overall, infections with *B. bassiana* had a greater impact on *T. ni* larvae immune induction compared to *C. javanica*. This is likely attributed to the higher virulence of *B. bassiana*. Whether additional AMPs are implicated in the defense response to *C. javanica* remains to be explored. Finally, while our results indicate different gene expression dynamics of the canonical immune pathways, it does not consider potential post-translational modifications that might affect the action of these immune signaling pathways. Further research (i.e., protein analysis or functional assays) is needed to confirm the contribution of these immune-related genes.

## 5. Conclusions

In conclusion, our study assessed the immune response of *T. ni* larvae to sublethal infections of two important entomopathogenic fungi. Five-day-old larvae were able to mount a robust response to fungal inoculation by exhibiting increased phenoloxidase activity, as well as upregulating expression across several key immune signaling pathways. These results, including the diverging larval responses to different fungal strains, provide insight into the immune strategies employed by *T. ni* for protection against entomopathogenic fungi. Moreover, these results increase our understanding of insect–fungal entomopathogen interactions and could aid in the development of more effective microbial control strategies for this ubiquitous pest of agricultural importance.

## Figures and Tables

**Figure 1 insects-14-00667-f001:**
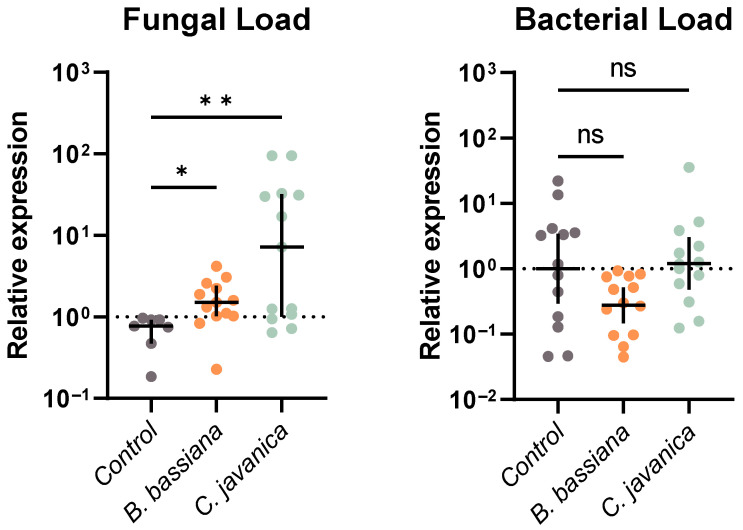
Fungal and bacterial loads via relative quantification of fungal 18s rRNA and bacterial 16s rRNA, respectively, in the whole bodies of *T. ni* larvae infected with either *B. bassiana* or *C. javanica*, evaluated at 2 days post-infection (sample size = 13 larvae/treatment). Each dot represents a single larva, with horizontal lines representing the median (fungal load) or mean (bacterial load) expression with interquartile range (fungal load) or 95% confidence intervals (bacterial load). * *p* < 0.05; ** *p* < 0.01; ns = not significant.

**Figure 2 insects-14-00667-f002:**
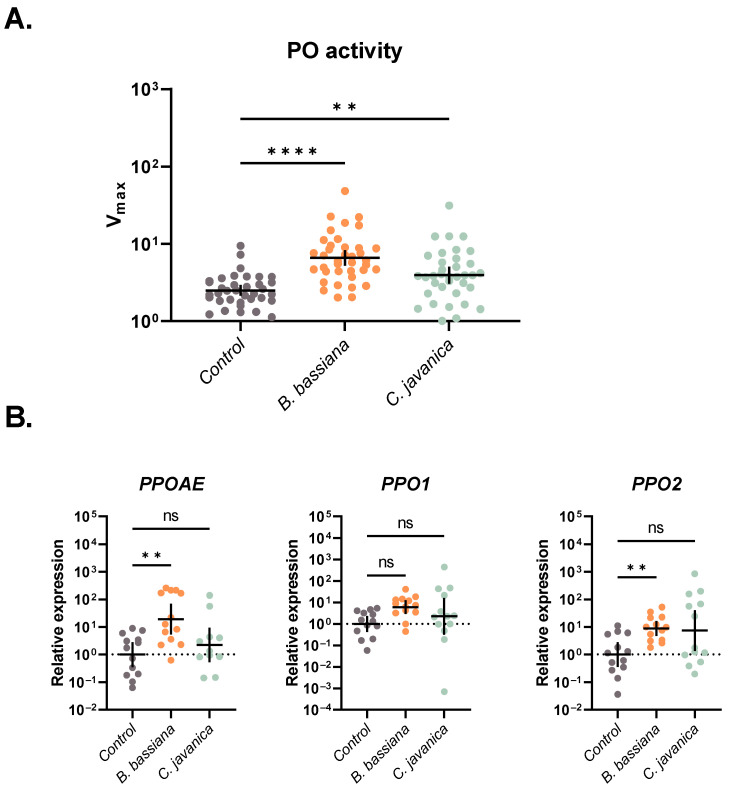
Gene expression profiles of (**A**) phenoloxidase activity (sample sizes: control = 36, *B. bassiana* = 38, *C. javanica* = 36) and (**B**) prophenoloxidase (sample size = 13 larvae/treatment) in *T. ni* infected with either *B. bassiana* or *C. javanica*, evaluated at 2 days post-infection. Each dot represents a single larva, with horizontal lines representing mean with 95% confidence intervals. ** *p* < 0.01; **** *p* < 0.0001; ns = not significant.

**Figure 3 insects-14-00667-f003:**
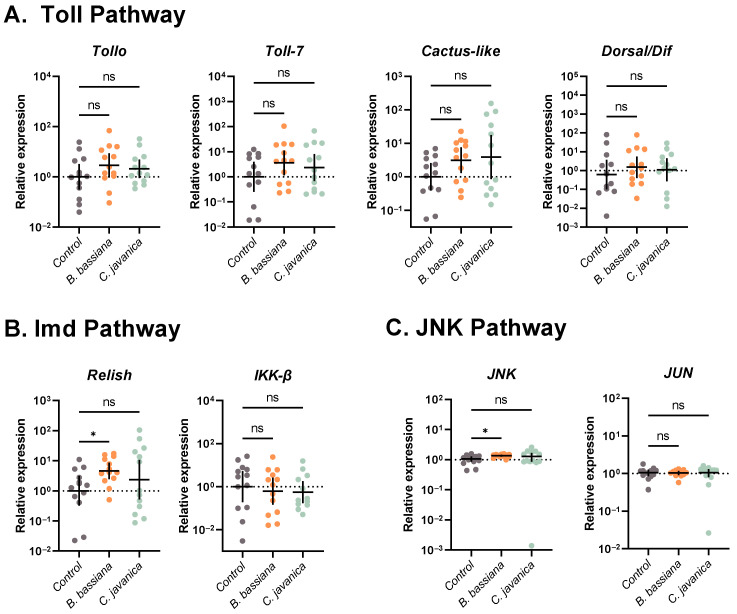
Gene expression profiles of (**A**) Toll, (**B**) Imd, and (**C**) JNK immune signaling pathways in *T. ni* infected with either *B. bassiana* or *C. javanica*, evaluated at 2 days post-infection (sample size = 13 larvae/treatment). Each dot represents a single larva, with horizontal lines representing mean expression with 95% confidence intervals. * *p* < 0.05; ns = not significant.

**Figure 4 insects-14-00667-f004:**
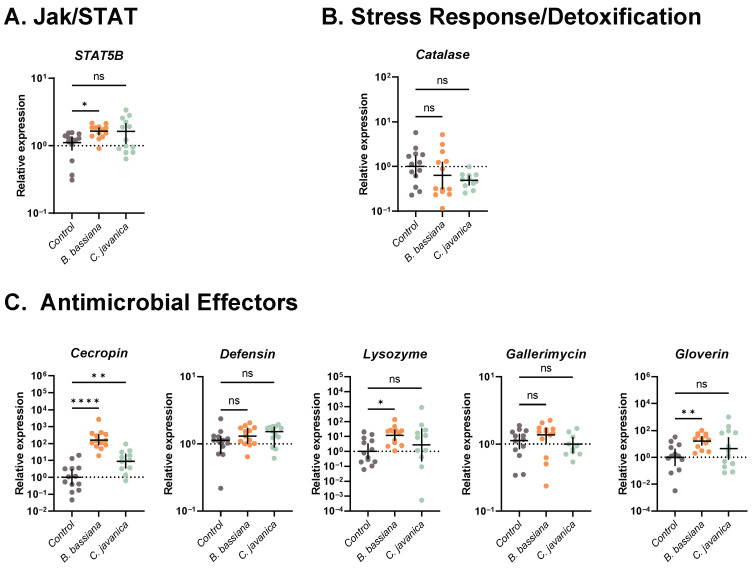
Gene expression profiles of genes involved in (**A**) Jak/STAT immune signaling pathway, (**B**) stress response/detoxification, and (**C**) antimicrobial effectors in *T. ni* infected with either *B. bassiana* or *C. javanica*, evaluated at 2 days post-infection (sample size = 13 larvae/treatment). Each dot represents a single larva with horizontal lines representing mean expression with 95% confidence intervals (*STATB* represents median and interquartile range). * *p* < 0.05; ** *p* < 0.01; **** *p* < 0.0001; ns = not significant.

**Table 1 insects-14-00667-t001:** Reverse Transcriptase Quantitative PCR (RT-qPCR) primers used to quantify gene expression in *Trichoplusia ni* larvae in this study.

Target	Gene ID	Primer Sequence	Reference
*Tollo*	XP_026733502.1	ACTCGACTGCAAAGCGAAAT	Present study
		TAAACCGTGTGGGGAACATT	
*Toll 7*	XP_026744547.1	ATCTCAAAACAGGGTCGCGT	
		AACACAGGTGATCGCTCAGG	
*NF-kappa-B inhibitor cactus-like*	XP_026732956.1	GAGCTTCTTGGTGACCTGCT	
		AGTGCTTACAAGCGCTGCTC	
*Dorsal/Dif*	XP_026732895.1	CCGCCTACAGGTTCCCTAAC	
		CCGGCGAGTACATCCTTTCA	
*Relish*	XP_026737640.1	CTCCCTTGATCAGGCACAAT	
		CCCTGAGGAACACCCTCATA	
*IKK-β*	XP_026741558.1	TGTTACAGACCTGCCGGAAC	
		CTTTGAGGCCCGAACAGGAT	
*STAT5B*	XP_026748140.1	GTGGACAATCACCACGACAG	
		CTGTTCCAGTCGCAGTTCAA	
*JUN*	XP_026743727.1	CGTGCTCCTTTAGCTTGACC	
		CTCCAAGCTGGAAGACAAGG	
*JNK*	XP_026741510.1	AACGTGTGAGGGGTTCGTAG	
		TGTTACCGCCCAGCTTTATC	
*Catalase*	XP_026730481.1	GACCTCAAAGATTCTCCTGGCT	
		AGAGCTGGTCCATTCTTGCC	
*Cecropin*	XP_026734190.1	TAGCCAAAATTGGAGCGAAG	
		AACCAGCTAGAGCGCCAATA	
*Defensin*	EU016385.1	CAATAAGCAGTGAAGCCTTGG	[37]
		GCATATGCCGTAGTTGTAGCC	
*Lysozyme*	EU016396.1	ATGCGCCAAGAAGATCTACAA	
		GTTTAGCATTTGCTGATGTCG	
*Gallerimycin*	EU016388.1	TGCATTGCCAGTTGTAGACAG	
		ATAGCCTCAAGCTCATCACCA	
*Gloverin*	EU016389.1	CTTGATGTCCACAAGCAGGTT	
		CAAAGGTCTTGTCCAGATTGC	
*ProPO activating enzyme (PPOAE)*	EU016397.1	AAGTCGGAAGAAGAGGTCGAG	
		CTGGCGTGTAACATGATCCTT	
*ProPO subunit 1 (PPO1)*	XP_026730296.1	GCTCCATGTATGCCAAGTGTT	Present study
		TTGGCCTTTCCATGAATGAT	
*ProPO subunit 2 (PPO2)*	XP_026731374.1	TCGTGGCGAACTTTTCTTCT	
		AGACTGTCCAGCTTCGGAAA	
*Ribosomal protein S18*	EU016398.1	TGTCCTATTTGTCGGGATGAG	[37]
		TGGTCCCATGCTCTTTCTATG	

## Data Availability

Data will be made available upon acceptance.

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
