# Peer review of "Increased Phenoloxidase Activity Constitutes the Main Defense Strategy of Trichoplusia ni Larvae against Fungal Entomopathogenic Infections"

_insects, 2023, doi:10.3390/insects14080667_

Round 1

Reviewer 1 Report

The cabbage looper, Trichoplusia ni, is an important agricultural pest almost all over the world. By RT-qPCR and PO activity assays, the authors have characterized the immune response of T. ni to two pathogenic fungal species, Beauveria bassiana (MBC 076) and Cordyceps javanica (MBC 439). These findings could provide the important insights for improving the biocontrol strategy of this pest. Collectively, this manuscript is well-organized and has clearly presented the results and their significance. So it can be accepted after minor revision.

Minor comments:

1. In the present manuscript, it is better if the authors clearly described the goal of each experiment before the introduction of methods and process in detail. These information will be very helpful for the understanding of the motivation for this research, the logical relationship across different experiments and the significance of  results. For instance, the reason for determining the fungal load and bacterial load after the exposure of pathogens should be told definitely.

2. L366: ....aimed at halting the spread of the [16], please make sure if there was anything that have missed here. 

Author Response

Reviewer 1)

The cabbage looper, Trichoplusia ni, is an important agricultural pest almost all over the world. By RT-qPCR and PO activity assays, the authors have characterized the immune response of T. ni to two pathogenic fungal species, Beauveria bassiana (MBC 076) and Cordyceps javanica (MBC 439). These findings could provide the important insights for improving the biocontrol strategy of this pest. Collectively, this manuscript is well-organized and has clearly presented the results and their significance. So it can be accepted after minor revision.

We thank the reviewer for his/her kind remarks.

Minor comments:

  1. In the present manuscript, it is better if the authors clearly described the goal of each experiment before the introduction of methods and process in detail. These information will be very helpful for the understanding of the motivation for this research, the logical relationship across different experiments and the significance of results. For instance, the reason for determining the fungal load and bacterial load after the exposure of pathogens should be told definitely.

We have added a few lines in each subsection throughout the methods section for added clarity.

  1. L366: ....aimed at halting the spread of the [16], please make sure if there was anything that have missed here.

We thank the reviewer for pointing out this error and have corrected it to say “Elicitation of immune signaling cascades during microbial infections result in the production of antimicrobial effectors, aimed at halting the spread of the spread of the infection [16].”

Reviewer 2 Report

Duffield et al investigated the immune response against two different entomopathogenic fungi to the cabbage looper. The level of immune response genes was checked by qRT-PCR after exposing the sublethal dose of the fungi. The manuscript was well-written and concise.

I am wondering why you chose the 2 days post infection larva. Did you try the earlier time points? It seems like different immune responses activate in different time points.

And second thought is that

Because the amount of transcripts increased, it can be seen that related genes were more involved in the pathway, but I think that Imd or JNK, which have increased slightly rather than quantitative manner, will cause a lot of aftermath in the pathway as well.

To sum up, an increase in the absolute amount of transcripts does not always mean an absolute contribution in pathways.

Minor comments:

Line 33 and 37: B. bassiana   - >  italic

Below the Figure, the figure legend should be attached the right bottom of the figure.

Line 350 and 355:  double space   make it to singe space after period.

Line 366: finish the sentence properly. 

Author Response

Reviewer 2)

Duffield et al investigated the immune response against two different entomopathogenic fungi to the cabbage looper. The level of immune response genes was checked by qRT-PCR after exposing the sublethal dose of the fungi. The manuscript was well-written and concise.

We thank the reviewer for his/her kind remarks.

I am wondering why you chose the 2 days post infection larva. Did you try the earlier time points? It seems like different immune responses activate in different time points.

The reviewer is correct, we would expect different immune responses being activated at different time points, as have been shown for other insect-fungal pathogen systems. Here, we chose 2-days post infection because preliminary work with this and other insects indicated limited and inconsistent gene regulation at earlier time points. This time point also coincides with the time at which blastospores start becoming more apparent and interacting with other insect cells in the hemocoel.

And second thought is that … because the amount of transcripts increased, it can be seen that related genes were more involved in the pathway, but I think that Imd or JNK, which have increased slightly rather than quantitative manner, will cause a lot of aftermath in the pathway as well. To sum up, an increase in the absolute amount of transcripts does not always mean an absolute contribution in pathways.

We agree with the reviewer. Our measurements of gene expression only evaluate transcript levels and does not take into account potential post-translational modifications or other regulatory mechanism governing the signaling pathways. We have added a paragraph indicating this statement in the discussion section.

New paragraph reads:

Lines 377-392: “Finally, while our results indicate different gene expression dynamics of the canonical immune pathways, it does not consider potential post-translational modifications that might affect the action of these immune signaling pathways. Further research (i.e., protein analysis or functional assays) is needed to confirm the contribution of these immune-related genes.”

Minor comments:

Line 33 and 37: B. bassiana   - >  italic

We have corrected these errors and throughout the manuscript.

Below the Figure, the figure legend should be attached the right bottom of the figure.

Although we chose different colored dots for each treatment, we have chosen to exclude a figure legend for each figure as axes are clearly labeled with each treatment. Adding a figure legend would not provide any new information or added clarity.

Line 350 and 355:  double space   make it to singe space after period.

These extra spaces have been deleted.

Line 366: finish the sentence properly.

This has been corrected. “Elicitation of immune signaling cascades during microbial infections result in the production of antimicrobial effectors, aimed at halting the spread of the spread of the infection [16].”